

# Up-regulation of Grb2-associated binder 1 promotes hepatocyte growth factor-induced endothelial progenitor cell proliferation and migration

Qing Fan[1,2], Liyu Zhang[1,2], Wenjie Zhu[1,2], Sheng Xue[3], Yisheng Song[1,2] and Qing Chang[1,2]

[1] Qingdao University, Qingdao, China
[2] Cardiovascular Surgery Department, The Affiliated Hospital of Qingdao University, Qingdao, China
[3] Institute for Translational Medicine, Qingdao University, Qingdao, China

## ABSTRACT

**Objectives:** Grb2-associated binder 1 (Gab1), a scaffolding adaptor protein, plays an important role in transmitting key signals that control cell growth, migration, and function from multiple tyrosine kinase receptors. This study was designed to investigate the influence of upregulation of Gab1 in endothelial progenitor cells (EPCs) stimulated with hepatocyte growth factor (HGF), and the underlying molecular mechanisms.

**Materials and Methods:** Endothelial progenitor cells isolated from human umbilical cord blood were identified and divided into four groups. EPCs in the Control group were cultured normally; those in the Control+HGF group were treated with HGF stimulation; those in the AD-Gab1 group were transfected with adenovirus containing the *Gab1* gene but not treated with HGF stimulation; and, those in the AD-Gab1+HGF group were treated with both HGF stimulation and transfection with adenovirus containing the *Gab1* gene. Subsequently, *Gab1* expression and proliferation and migration ability were compared for EPCs grown under different conditions. Furthermore, we measured phosphorylation levels of three key proteins Gab1, SHP2, and ERK1/2.

**Results:** The AD-Gab1+HGF group had the highest expression of Gab1 and higher proliferation and migration than the other three groups.

**Conclusions:** Upregulation of Gab1 promoted HGF-induced EPC proliferation and migration. Mechanistically, HGF stimulated Gab1 tyrosine phosphorylation in EPCs, thus leading to activation of extracellular regulated MAP kinase 1/2, which is involved in proliferation and migration signaling.

## INTRODUCTION

Heart and cardiovascular-associated conditions are major causes of death worldwide, and the number of people affected is continually increasing. According to World Health Organization statistics, 17.9 million deaths were due to cardiovascular disease in 2016,

Corresponding author
Qing Chang,
changqing20671@163.com

accounting for 44% of mortality worldwide (*World Health Statistics (WHO), 2018*). Heart valve disease makes up a large proportion of cardiovascular conditions and affects more than 100 million people worldwide (*Members et al., 2015*).

Currently, mechanical or biological prostheses are the "gold standard" treatment for heart valve failure (*Baumgartner et al., 2017*). However, both types of valves cannot grow and regenerate—functions particularly important in congenital heart defects, and patients often require subsequent reoperation as they age (*Blum, Drews & Breuer, 2018*). All available prostheses have other disadvantages, such as a risk of infection, constant anticoagulation in mechanical valves, progressive degeneration in bioprostheses, and lack of a tissue source in homografts (*Best et al., 2016*). The tissue-engineering heart valve (TEHV) can potentially overcome most of the current shortcomings by providing a viable valve capable of growth, remodeling, and repair. The ultimate goal of TEHV is to produce a structure similar to the native valve structure (*Emmert et al., 2014*). In 2006, *Cebotari et al. (2006)* reported two successful clinical applications of TEHV in humans by using autologous endothelial progenitor cells (EPCs). As one of the main cell sources of TEHV, EPCs can attach, multiply, and cover the implant with an endothelial layer. However, autologous EPCs are usually insufficient in patients who require heart valve replacement, and the covering with an endothelial layer occurs slowly (*Rippel, Ghanbari & Seifalian, 2012*; *Bianconi et al., 2018*). The absence of this autologous endothelial layer may cause immunological reactions. If autologous EPC proliferation and migration could be enhanced, the re-endothelialization time—that is, the time required for TEHV to form an endothelial layer—could be decreased, and the immunological reactions could be alleviated.

The Grb2-associated binder (Gab) family docking proteins are involved in the amplification and integration of signal transduction evoked by growth factors, cytokines, antigens, and numerous other molecules (*Nakaoka & Komuro, 2013*). Gab1 has an essential role in postnatal angiogenesis and arteriogenesis via hepatocyte growth factor (HGF)/c-Met signaling. It is associated with SHP2 after stimulation with HGF, which is required for activation of ERK1/2 (*Aasrum et al., 2013*). Downregulation of Gab1 inhibits cell proliferation and migration (*Sang et al., 2013*; *Xu et al., 2018*). In addition, Gab1 is required for HGF-induced endothelial cell (EC) proliferation and migration (*Zhao et al., 2011*). We hypothesized that Gab1 might play the same role in EPCs, which are EC precursors, and that upregulation of Gab1 in EPCs might have a positive effect on proliferation and migration.

Here, we demonstrated that upregulation of Gab1 promotes HGF-induced EPC proliferation and migration.

## MATERIALS AND METHODS

### Isolation and culture of EPCs

Compared with adult peripheral blood or bone marrow progenitors, cord blood progenitors have distinct proliferative advantages, and cord blood can be obtained non-invasively (*Murohara et al., 2000*). Human umbilical cord blood was collected from the placental cords of volunteers undergoing Cesarean section delivery. Collection

occurred immediately after the delivery of the placenta to avoid clot formation. Seven placental cords were utilized in this study. The EPCs were isolated and cultured as described previously (*Murohara et al., 2000*; *Li et al., 2016*). Briefly, mononuclear cells were isolated from the human cord blood by density gradient centrifugation over Histopaque-10771 (Sigma-Aldrich, St. Louis, MO, USA), according to the manufacturer's protocol. The cells were seeded in T25 flasks pre-coated with 0.1 mg/ml of human fibronectin (Sigma-Aldrich, St. Louis, MO, USA) and were incubated in EGM-2 BulletKit medium (Lonza, Cologne, Germany). After 3 days, non-adherent cells were removed, and the medium was replaced. Subsequently, the medium was changed every 2 days.

## Characterization of EPCs

After 7 days of culture in vitro, the EPCs were characterized as adherent cells, which were double-positive for acetylated low-density lipoprotein (acLDL) uptake and lectin binding, as assessed by direct fluorescent staining, as described previously (*Roberts et al., 2007*).

Briefly, to evaluate the ability of acLDL uptake and lectin binding in EPCs, the cells were cultured in 10 μg/mL of 1, 1′-dioctadecyl-3, 3, 3′, 3-tetramethyl-indocarbocyanine perchlorate-labeled acLDL (Sigma-Aldrich, St. Louis, MO, USA) for 4 h at 37 °C. They were then fixed with 2% paraformaldehyde for 15 min. The cells were washed with PBS and reacted with fluorescein isothiocyanate (FITC)-labeled *Ulex europaeus* agglutinin-1 (UEA-1, 10 μg/mL; Sigma-Aldrich, St. Louis, MO, USA) at room temperature for 1 h. The cells were washed to remove the free UEA-1. Nuclear counterstaining was performed with DAPI (4′, 6-diamidino 2-phenylindole; Sigma-Aldrich, St. Louis, MO, USA), and the cells were examined under a fluorescence microscope (Nikon, Tokyo, Japan). The adherent cells that stained with triple-positive fluorescence were determined to be EPCs. Nuclear staining with DAPI verified that nearly all the adherent cells (>95%) were acLDL (+) ulex-lectin (+).

Expression of endothelial lineage surface markers was evaluated by flow cytometry (BD Accuri C6; BD Biosciences, Sparks, MD, USA) using PE mouse anti-human CD31 antibody (BioLegend, San Diego, CA, USA), PE anti-human KDR antibody (BioLegend, San Diego, CA, USA). Flow cytometry detection of hematopoietic cells was performed using antibodies against hematopoietic cell-specific surface antigen such as PE anti-human CD133 antibody (BioLegend, San Diego, CA, USA). Appropriate isotype control was PE Human IgG1 Isotype control recombinant antibody (BioLegend, San Diego, CA, USA). Data were analyzed by using the BD Accuri C6 software package (BD Biosciences).

## Adenovirus transfection

AD-Gab1 was purchased from Biowit Technologies (Shenzhen, China). EPCs were harvested with 0.25% trypsin-EDTA (Gibco, Carlsbad, CA, USA) plated and seeded on 96-well plates at a density of 8,000 cells per well. After 24 h, 100 μL AD-Gab1 medium at six different multiplicity of infection (MOI) levels was added into each well for EPC transfection in triplicate. After 24 h, the cells were examined under a fluorescence microscope, and fluorescence intensities were measured in ImageJ. A suitable MOI was

determined and used for subsequent EPC transfection with AD-Gab1. After the cell intensity reached approximately 80%, RNA and protein were extracted from cells. We used a PrimeScript RT Reagent Kit (Takara, Dalian, China) to synthesize cDNA, then examined the expression of Gab1 by using quantitative real-time PCR with SYBR Premix Ex Taq II (Takara, Dalian, China), in triplicates run three times each. For Gab1, we used the forward primer 5′-TGCCATTAACTGTGCTTCCCA-3′ and the reverse primer 5′-TCGCACAGAGCACTCCAAAT-3′. For β-actin, we used the forward primer 5′-CTCCATCCTGGCCTCGCTGT-3′ and the reverse primer 5′-GCTGTCACCTTCACC GTTCC-3′. The relative Gab1 expression was calculated in Bio-Rad CFX Manager. Protein was prepared for western blot analysis in triplicate.

## Measurement of cell proliferation

Endothelial progenitor cells that were cultured at 37 °C and 5% $CO_2$ in an incubator for 14 days were harvested with 0.25% trypsin-EDTA and plated onto 96-well plates with EGM-2 (contain 5% FBS). The cells were incubated at 37 °C and 5% $CO_2$ for 24 h. We added 0.5 μL of 50 μM EdU (Sigma-Aldrich, St. Louis, MO, USA) into each well containing 500 μL of medium and incubated the cells for 4 h. The cells were fixed with 4% paraformaldehyde and incubated with two mg/mL aminoacetic acid for 5 min with oscillation. The cells were incubated with 100 μL of the penetrant into each well and oscillated 10 min, and 100 μL of 1 × EdU solution was then added and incubated for 30 min. DAPI was used to stain cell nuclei.

After digestion with pancreatic enzyme, EPCs were plated on 96-well plates with a density of approximately 4,000 cells per well, in triplicate. The next day (24 h later), the AD-Gab1 group and AD-Gab1+HGF group were transfected with AD-Gab1 at MOI = 20. One plate was randomly chosen to run CCK-8 (7Sea Pharmatech, Shanghai, China) tests, to determine 24 h proliferation. The medium of the Control+HGF group and AD-Gab1+HGF group was changed to EGM-2 with 5% FBS and 20 ng/mL of HGF (PeproTech, Rocky Hill, NJ, USA), whereas the Control group and AD-Gab1 group were maintained in EGM-2 with 5% FBS and without HGF. The cells were incubated for another 24 h, and cell proliferation was then tested (72 h time point). Subsequently, we removed HGF stimulation in Control+HGF group and AD-Gab1+HGF group, performed CCK-8 tests at 96 and 120 h with the last two plates.

## In vitro wound-healing assay

The cells were removed by trypsinization, counted, and plated at a density of $1 \times 10^6$ cells/per well in six-well plates. After transfection, cells were incubated until confluent monolayers were formed for wounding assays. Wounds were made with a pipette tip, and photographs were taken immediately (time 0). Then the medium of the Control+HGF group and AD-Gab1+HGF group was changed to EGM-2 without FBS and with 20 ng/mL of HGF; the Control group and AD-Gab1 group were maintained in EGM-2 without FBS. Photographs were taken at 12, 24, 36, and 48 h after wounding. The cell-covered area (%) was measured to determine the amount of migration by the cell monolayer to cover the wounded area during this time period. The cell-covered area (%)

was determined as $(W_0 - W_t)/W_0$, where $W_t$ represents the wound area (with no cells) at time $t$. These areas were measured and analyzed in ImageJ.

## Western blot analysis

Cells were harvested in RIPA buffer containing a protease inhibitor cocktail. Total protein concentration was determined using a Bradford protein assay kit (Solarbio, Beijing, China). Supernatant and lysis samples were separated with SDS-PAGE (8% or 10%) and transferred onto a polyvinylidene difluoride membrane, blocked with 5% non-fat dry milk and probed overnight at 4 °C with appropriate primary antibodies (Cell Signaling Technology, Danvers, MA, USA). After washing and incubation with secondary antibodies (Cell Signaling Technology, Danvers, MA, USA) and visualized using Power-Opti ECL™ solution (Millipore, MA, USA) and a cooled CCD camera system (Vilber Fusion Solo 4S; Paris, France).

## Statistical analysis

The original data were processed in GraphPad Prism 7.0. The quantitative real-time PCR data were analyzed with an independent-samples $t$-test. The western blot analysis in Fig. 1 was performed with a Student's $t$-test. Other statistical analysis was performed with ordinary one-way ANOVA. All these tests were performed on at least three independent biological replicates. All data are expressed as the mean ± SD. A value of $p < 0.05$ was considered statistically significant.

## Ethics approval and consent to participate

We have received informed consent from participants. All the procedures were followed by the Medical Ethics Committee of The Affiliated Hospital of Qingdao University, Qingdao.

# RESULTS

## Characterization of EPCs

Freshly isolated mononuclear cells were the small round cells suspended in the medium. With increased culturing time, the cells gradually stretched and became larger, forming cell–cell adhesions. Cell colonies were observed at approximately 7 days after seeding. A total of 3–4 weeks later, the ECs showed a typical cobblestone-like appearance. After 7 days' culture in vitro, when observed under a fluorescence microscope, the adherent dil-acLDL-labeled cells were red, the cells bound with FITC-UEA-I were green, and the cell nuclei were stained blue with DAPI. Almost 100% of the cells showed three colors, and these cells were considered the differentiating EPCs (Figs. 1A–1F). The cells were positive for CD31, KDR and negative for hematopoietic cell surface antigen CD133 (Fig. 1K) which is in accordance with previously published data (Ingram et al., 2004).

## Adenovirus transfection into EPCs

The efficiency of adenovirus transfection was determined by fluorescence microscopy. The more EPCs transfected, the more green fluorescence was observed. An excessive MOI may damage EPCs and even lead to cell death. The statistical results indicated that, at MOI = 20, the number of cells with green fluorescence was much greater than at other

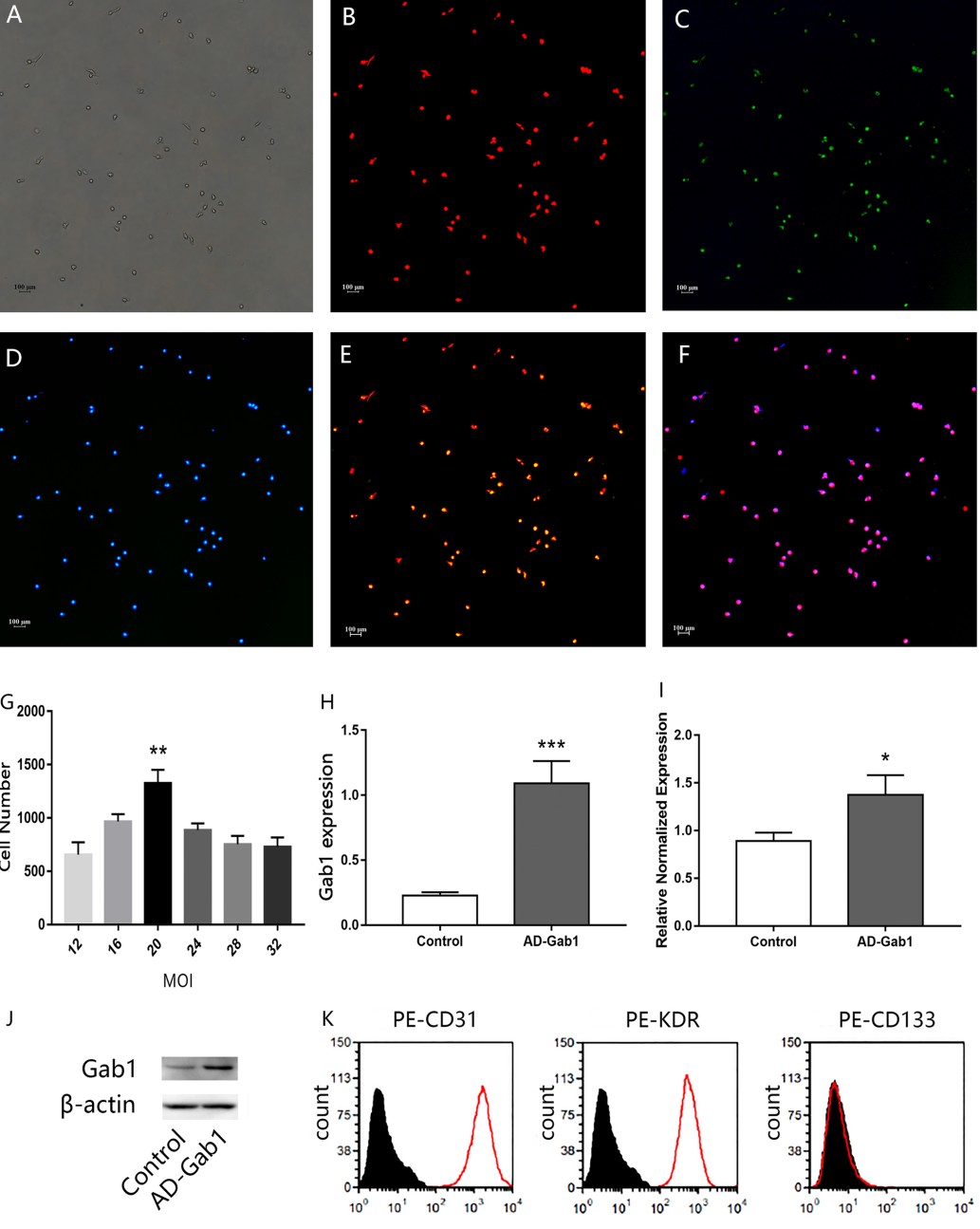

**Figure 1 Characterization of EPCs.** Immunofluorescence staining of EPCs was performed after 7 days (orig. mag. × 40). (A) Unstained EPCs; (B) dil-acLDL-labeled cells (red); (C) FITC-UEA-I-bound cells (green); (D) DAPI contrast staining of the cell nuclei; (E) dil-acLDL-labeled and FITC-UEA-I-bound double positive cells; (F) dil-acLDL-labeled, FITC-UEA-I-bound, and DAPI stained triple positive cells. Fluorescence indicating Gab1 expression in EPCs transfected with adenovirus (24 h) with different multiplicity of infection (MOI) levels, and expression of Gab1 at the optimal MOI. (G) The number of fluorescent cells at different MOI levels. (H) qPCR results show the expression of the Gab1. (I, J) The expression of the Gab1 after transfected (western blot). (K) Flow cytometry characterization of EPCs for CD31, KDR, and CD133. Plots depict control isotype IgG staining (black histograms) vs specific antibody staining (empty histograms). $^{**}p < 0.01$ ($n = 3$) vs other MOI groups, $^{*}p < 0.05$ ($n = 3$) vs Control, $^{***}p < 0.001$ ($n = 3$) vs Control. Values are the mean ± SD.

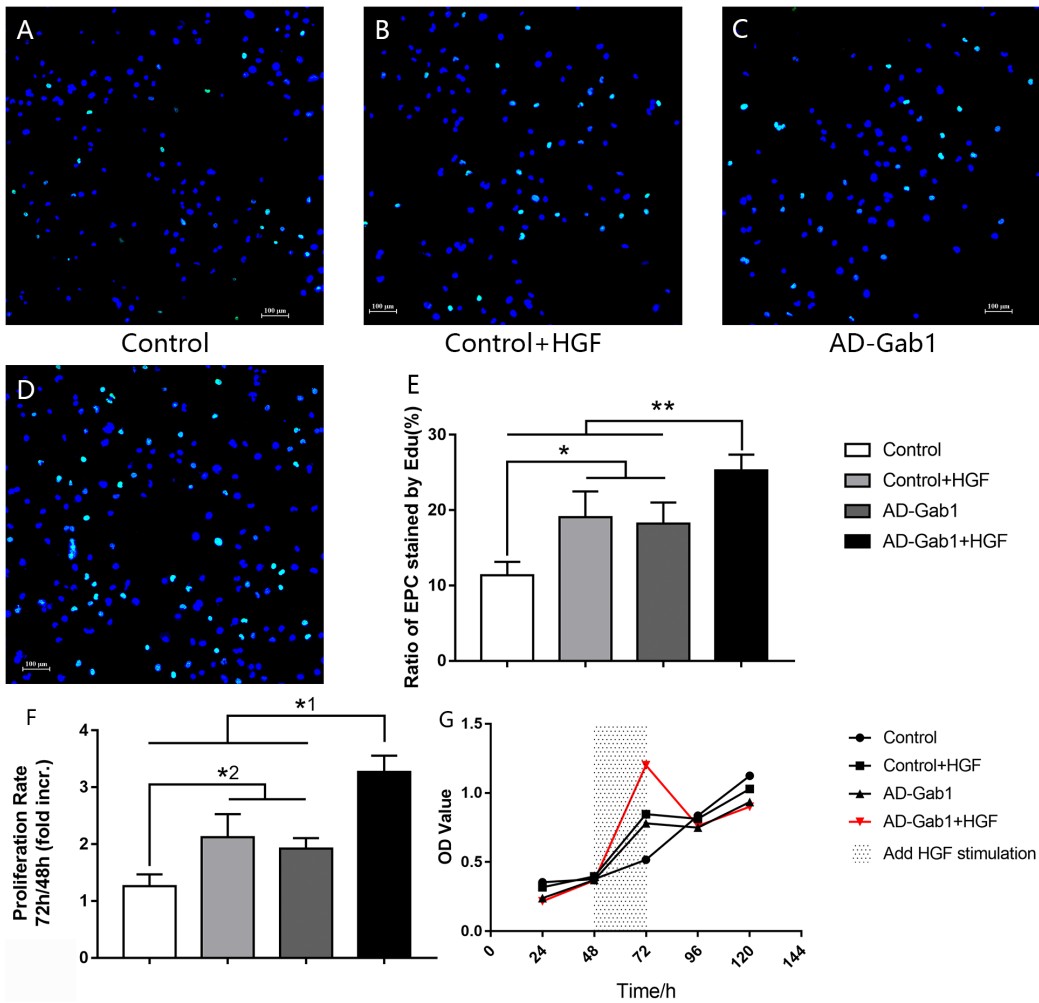

**Figure 2 EdU incorporation reflecting the proliferation of EPCs.** EdU incorporation reflecting the proliferation of EPCs cultured for 14 days. (A–D) DAPI (blue) was used to stain nuclei, and EdU (green) was incorporated into EPCs in each group (orig. mag. × 100). (E) The ratio of EPCs stained by EdU in each group. $^*p < 0.05$ ($n = 5$) vs Control+HGF and AD-Gab1 group, $^{**}p < 0.01$ ($n = 5$) vs Control, Control+HGF and AD-Gab1 group. CCK-8 cell viability tests reflecting the proliferation of EPCs in different groups. (F) Cell proliferation in each group. (G) Growth curves of EPCs in different time and treatment conditions. $^{*1}p < 0.05$ ($n = 4$) vs Control, Control+HGF and AD-Gab1 group. $^{*2}p < 0.05$ ($n = 4$) vs Control+HGF and AD-Gab1 group. Values are the mean ± SD.

MOI levels (Fig. 1G). After we selected the proper MOI, we sought to validate the adenovirus transfection into EPCs. The qPCR results showed that the expression of the *Gab1* gene was upregulated in the AD-Gab1 group after transfection (Fig. 1I), and western blot analysis demonstrated that Gab1 protein expression was higher in the AD-Gab1 group (Figs. 1H and 1J).

## Effect of overexpression of Gab1 on EPC proliferation

Detection of EdU is more sensitive than BrdU detection and can be accomplished in minutes (*Salic & Mitchison, 2008*). Under a fluorescence microscope, we observed EdU

incorporation to determine the proliferation of EPCs that were cultured for 14 days (Figs. 2A–2D). The proliferation of EPCs in the Control+HGF group and AD-Gab1 group was increased, whereas the proliferation in the AD-Gab1+HGF group was even higher, and the results were statistically significant (Fig. 2E).

In CCK-8 tests, the optical density (OD) was measured at 450 nm wavelength. The greater the cell number, the higher the OD. Because we added HGF stimulation between 48 and 72 h, the cell proliferation rate was determined as the OD value at 72 h divided by the OD value at 48 h. Statistical analysis of the data confirmed the results of the EdU incorporation assay. The cell proliferation ability of the Control+HGF group and AD-Gab1 group was higher than that of the Control group between 48 and 72 h (Fig. 2F). The cell proliferation ability of the AD-Gab1+HGF group was significantly different from that of the other three groups (Fig. 2F). We plotted growth curves with the mean OD values as the ordinate and the cell culture time as the abscissa (Fig. 2G). After HGF stimulation, EPCs in the AD-Gab1+HGF group grew most rapidly.

In conclusion, HGF-stimulation and Gab1 have a role in promoting cell proliferation, and upregulation of Gab1 expression enhances this effect.

## Effect of overexpression Gab1 on EPC migration

In wound-healing assays, we determined the area covered by a wounded cell monolayer on plastic after different treatments. When we used EGM-2 without FBS, EPCs had little proliferation ability. The results are reported as cell-covered area (Figs. 3A and 3B). The cell-covered areas in the Control+HGF group and AD-Gab1 group were higher than that in the Control group, and the increase in cell-covered area in the Control+HGF group was higher than that in the AD-Gab1 group. The data suggested that HGF stimulation and upregulation of Gab1 both influence EPC migration. HGF stimulation had a greater effect on EPC migration than up-regulation of Gab1 expression. We presumed that HGF stimulation combined with upregulation of Gab1 to treat EPCs should result in a greater improvement in EPC migration. To test this possibility, we analyzed an AD-Gab1+HGF group. At 12 h, the cell-covered area in the AD-Gab1+HGF group was significantly higher than those in the other three groups (Fig. 3A). At 24, 36, and 48 h, the cell-covered area in the AD-Gab1+HGF group was higher than those in the other groups (Fig. 3A) and increased more quickly than those in the other groups (Fig. 3C). Thus, EPCs treated with both HGF stimulation and transfection of adenovirus containing the Gab1 gene had a greater migration ability.

## HGF-mediated signaling in EPCs

The ERK signaling pathway is involved in cell proliferation and migration. To investigate how upregulation of Gab1 mediates HGF-induced EPC proliferation and migration, we assessed the activation of ERK1/2 in response to HGF stimulation and upregulation of Gab1 expression. The expressions of c-Met, Gab1, phospho-Gab1, SHP2, phospho-SHP2, ERK1/2, and phospho-ERK1/2 in EPCs was analyzed via western blotting. The expression of the c-Met in AD-Gab1+HGF group was consistent with the other groups (Fig. 4E). Quantitative analysis demonstrated that the phosphorylation of

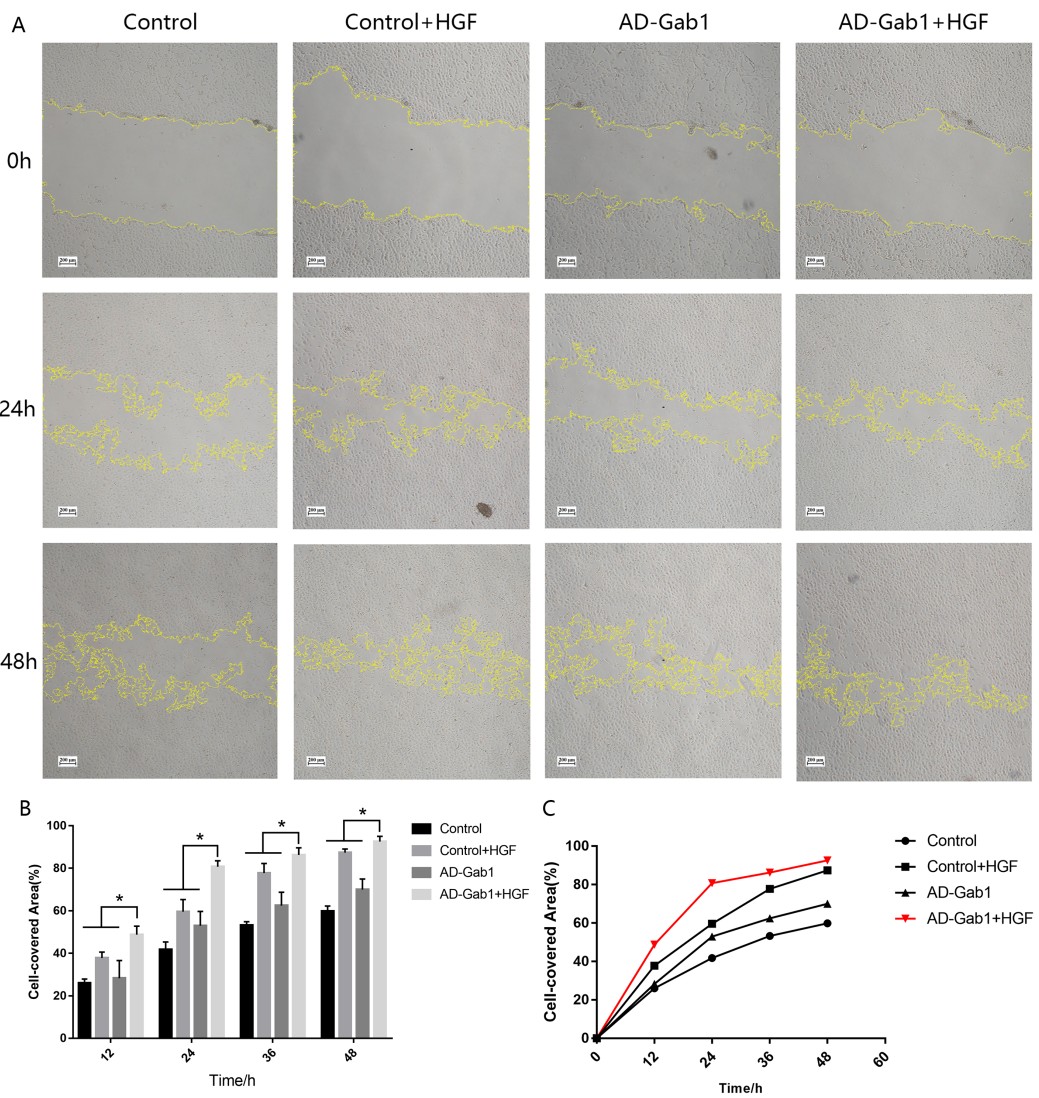

**Figure 3 Wound-healing assays reflecting the migration of EPCs in different groups.** (A) Wound-healing assays reflecting the migration of EPCs in different groups. (B) Cell-covered area in each group at different times. (C) Cell-covered area increased over time. $^*p < 0.05$ ($n = 3$) vs Control, Control+HGF and AD-Gab1 group. Values are the mean ± SD.     

Gab1 in the AD-Gab1 group was almost equal to that in the Control group but lower than that in the Control+HGF group and AD-Gab1+HGF group (Fig. 4B). We confirmed that HGF stimulated Gab1 tyrosine phosphorylation, as previously reported (*Zhao et al., 2011*). Once sufficient Gab1 expression was achieved, for example in the AD-Gab1+HGF group, HGF induced more phosphorylation of Gab1 (Fig. 4B). The increased Gab1 phosphorylation substantially increased the levels of SHP2 phosphorylation, thus leading to activation of MAP kinase 1/2, which is involved in proliferation and migration signaling (Figs. 4C and 4D). Together, these results established a critical role of EPCs Gab1 in proliferation and migration signaling and indicated that upregulation of Gab1 enhances the function of this signaling pathway.

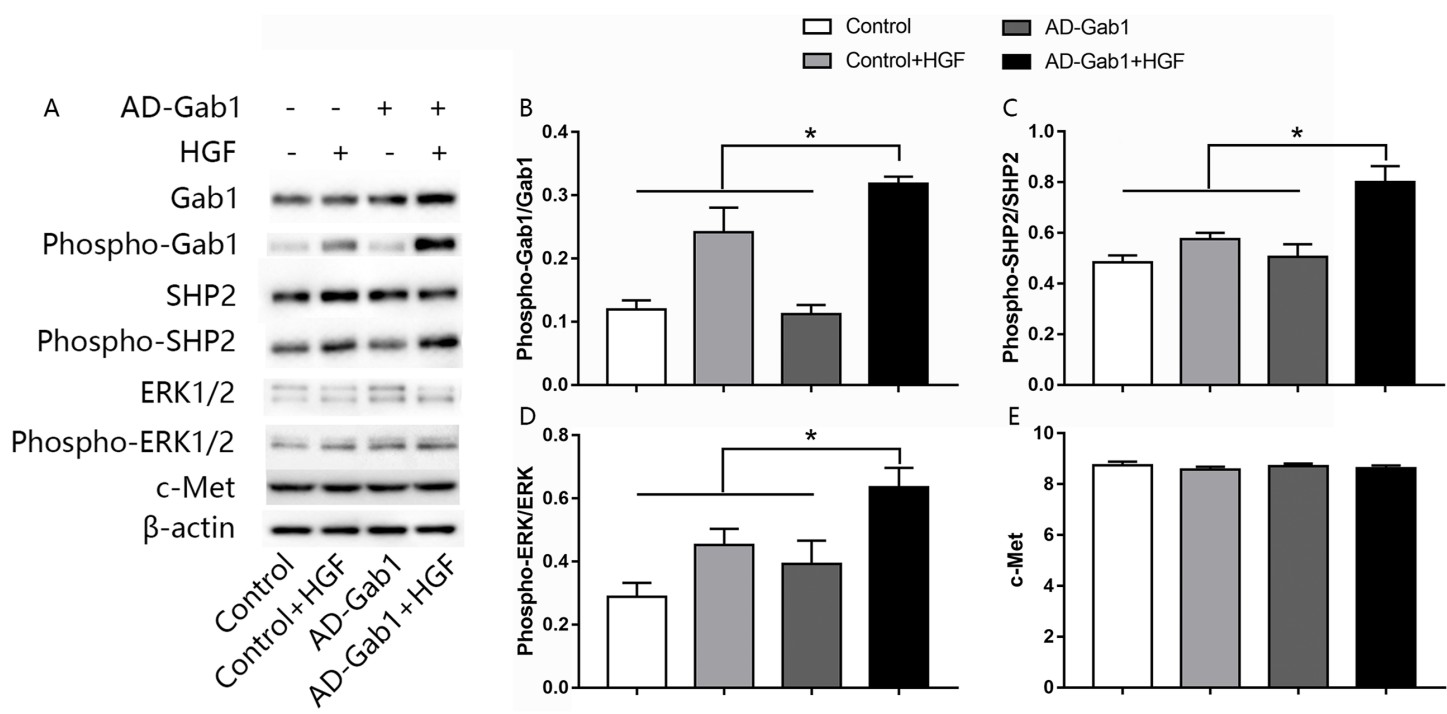

**Figure 4 Effect of overexpression of Gab1 on HGF-induced phosphorylation of SHP2 and ERK1/2.** The expression of c-Met, the phosphorylation of SHP2 and ERK1/2 was determined by western blot. Representative blots (A) and bar graphs summarizing the expression of c-Met (C) and the effects of the overexpression of the Gab1 (B) on the HGF-induced (20 ng/mL) phosphorylation of SHP2 (C), and ERK1/2 (D). $*p < 0.05$ ($n = 3$) vs Control, Control+HGF and AD-Gab1 group. Values are the mean ± SD.

## DISCUSSION

*Asahara et al. (1997)* first reported that a subtype of hematopoietic progenitor cells from adults, namely EPCs, have EC features and can differentiate into mature ECs and might represent a potential strategy for TEHV. Subsequently, experiments confirmed that TEHV using autologous EPCs is very encouraging (*Cebotari et al., 2006*; *Dohmen et al., 2007*; *Sales et al., 2010*). However, EPCs are a fairly rare cell population, and, when administered intravenously, only a very small fraction of EPCs reach the target region and participate in re-endothelialization. How to increase their cell number is an urgent problem.

Many receptor tyrosine kinases (RTKs), including vascular endothelial growth factor (VEGF), are shown to regulate EC function (*Limbourg et al., 2009*). Except VEGF, HGF is an angiogenic factor which stimulates EC proliferation and migration (*Eric et al., 2005*; *Bonauer et al., 2009*). HGF binds its receptor c-Met and stimulates c-Met kinase activation, which triggers transphosphorylation of c-Met and downstream signaling events (*Ha et al., 2008*). The HGF/c-Met pathway has emerged as a promising therapeutic target by which to promote re-endothelialization in vivo.

Unlike most RTKs recruit Gab1 indirectly via Grb2, Gab1 can be recruited to activated c-Met through direct mechanism, interacts with tyrosine-phosphorylated c-Met via the Met-binding domain (amino acids 450–532) (*Gu & Neel, 2003*; *Barrow-McGee & Kermorgant, 2014*). Mechanistically, we found that Gab1 is critical in mediating

HGF/c-Met proliferation and migration signaling of human EPCs in vitro. To our knowledge, this is the first study revealing a key role of upregulation of Gab1 in promoting proliferation and migration and HGF-mediated signaling in human EPCs.

Previous studies on the role of Gab1 have usually used a Gab1-ecKO mice model. Those studies have found that Gab1-deficient mice have developmental defects in postnatal angiogenesis (*Xu et al., 2018*; *Wang et al., 2015*). However, the effect of upregulating Gab1 in human EPCs has not been explored. Using human umbilical cord blood, we isolated human EPCs and upregulated their Gab1 expression through adenovirus transfection. We showed that upregulation of Gab1 markedly enhanced the proliferation and migration of human EPCs. HGF has been shown to mobilize and increase EPC number (*Rehman et al., 2004*). Our results showed that under the same dose of HGF stimulation, the proliferation and migration of human EPCs was strongly enhanced in EPCs in which Gab1 was upregulated. Therefore, upregulation of Gab1 amplifies stimulation by HGF.

The results also supported that Gab1 is important for HGF-induced ERK1/2 phosphorylation. Few studies in other cells, such as HUVEC and MDCK cells, have evaluated the role of Gab1 in HGF-induced ERK1/2 (*Maroun et al., 2000*; *Shioyama et al., 2011*; *Aasrum et al., 2015*). In these cells, which were transfected to express mutated Gab1 unable to recruit SHP2, the HGF induced sustained activation of the ERK pathway was found to be reduced (*Maroun et al., 2000*). ERK1/2 activation has been shown to regulate cell migration and survival signaling pathways (*Koch et al., 2011*). Gab1 plays an important role in mediating growth factor-induced activation of ERK1/2 through recruiting SHP2 in a tyrosine phosphorylation-dependent manner (*Aasrum et al., 2015*). However, the role of Gab1 in HGF mediated signaling in human EPCs remains unclear. Our results suggested that upregulation of Gab1 in human EPCs results in recruitment of more SHP2. We further found that under the same dose of HGF stimulation, greater SHP2 tyrosine phosphatase activation and ERK1/2 phosphorylation were observed in cells overexpressing Gab1. The data suggest signaling mechanisms by which Gab1 mediates growth factor-induced proliferation and migration.

## CONCLUSIONS

In this study, we found that upregulation of Gab1 promotes HGF-induced EPC proliferation and migration. Mechanistically, HGF stimulates Gab1 tyrosine phosphorylation in EPCs, thus leading to activation of extracellular regulated MAP kinase 1/2, which is involved in proliferation and migration signaling. Our findings may have clinical implications: they suggest that enhancing Gab1 signaling may be a potential strategy to increase EPC cell number and provide a new means of achieving rapid TEHV re-endothelialization.

## ACKNOWLEDGEMENTS

We wish to thank all colleagues at the Institute for Translation Medicine of Qingdao University for help during all phases of the project.

### Funding

This study was supported by the Science and Technology Development Program, Shandong Province, China (No. 2013GHY11504). The funders had no role in study design, data collection and analysis, decision to publish, or preparation of the manuscript.

### Grant Disclosure

The following grant information was disclosed by the authors:
Science and Technology Development Program, Shandong Province, China: 2013GHY11504.

### Competing Interests

The authors declare that they have no competing interests.

### Author Contributions

- Qing Fan conceived and designed the experiments, performed the experiments, analyzed the data, contributed reagents/materials/analysis tools, prepared figures and/or tables, authored or reviewed drafts of the paper.
- Liyu Zhang performed the experiments.
- Wenjie Zhu performed the experiments.
- Sheng Xue conceived and designed the experiments.
- Yisheng Song performed the experiments, contributed reagents/materials/analysis tools, providing jokes.
- Qing Chang authored or reviewed drafts of the paper, approved the final draft.

### Human Ethics

The following information was supplied relating to ethical approvals (i.e., approving body and any reference numbers):

The Medical Ethics Committee of The Affiliated Hospital of Qingdao University granted Ethical approval to carry out the study within its facilities.

### Data Availability

Raw data is available in the Supplemental Files.

### Supplemental Information

Supplemental information for this article can be found online at http://dx.doi.org/10.7717/peerj.6675#supplemental-information.

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
