# Peer review of "Up-regulation of Grb2-associated binder 1 promotes hepatocyte growth factor-induced endothelial progenitor cell proliferation and migration"

_PeerJ, doi:10.7717/peerj.6675_

## Round 0.1 · original submission · Major Revisions

Please respond to reviewers comments, particularly where clarification is requested. Some discussion of pTyr sites on Met receptor vs. other RTKs may be helpful as a means of explaining the particular link to Met receptor of Gab1 as a signaling scaffold. The contrast of wound healing images needs to be increased so that cells are more clearly discernible. I share some of the misgivings of reviewer 2 concerning the physiological relevance of this study.

·

Basic reporting

The text is clear throughout without need for correction.

For the most part, there is sufficient background. However, on lines 93-95, you mention the fluorescent staining assessment of EPCs. You end with "...as described previously". Please supply a reference for this statement.

Experimental design

In the methods section (lines 153-161), the authors describe both immunoprecipitation and western blotting, yet it is unclear where the IP was performed and to what purpose. This needs to be clarified as it appears to me from the figures presented that only western blot was performed.

On what basis were SHP and ERK1/2 chosen as key downstream indicators of Gab1 signalling? Was PI3K/Akt Which residues of the target signalling proteins were the phospho-specific antibodies directed towards?

Why was 20ng/ml HGF chosen? The amount of ligand has a significant effect on the cellular readout. Has this concentration been optimised previously (if so, can this be referenced)?

EPC migration was performed in EGM-2 without FBS. It is unclear whether earlier experiments assessing proliferation were also performed without FBS. This should be clarified. If there is a difference between the two methods, this should be explained.

Did the authors assess the level of the MET receptor on the EPCs? As this is the major receptor through which HGF signals, it would be useful to ensure that this is consistent between the groups.

EGM-2 medium contains a number of different growth factors pertinent to endothelial cell growth (e.g. VEGF, EGF, FGF etc.). Many of the receptor tyrosine kinases through which these growth factors signal also utilise Gab1 (as mentioned by the authors). Overexpression of Gab1 in the absence of HGF increases proliferation to a similar extent to control + HGF. Can the authors provide an explanation as to why HGF has a much larger effect than all other Gab1-related growth factors combined?

At 96 and 120 hours, the OD values have dropped back significantly so that the AD-Gab1+HGF is no longer different to the other conditions. If the OD represents a surrogate marker of the increase in the number of cells over time, then this suggests a significant loss of cells between 72 and 96 hours. Can the authors provide some explanation as to what has happened? Is this due to overproliferation and therefore lack of nutritional support in the remaining media? Promotion of proliferation is a major outcome measure in the study but it must be sustainable and the new cells must be healthy if it is to have any benefit for TEHV.

Validity of the findings

I have no problems with the validity of the findings, although further clarity is needed in areas mentioned above.

More information is required about the samples to determine the robustness of the method. How many cords were utilised in the study? In the statistical analysis section it is stated that "All these tests were performed on at least three replicates." - were these technical replicates or independent biological replicates from n=3 different cords? This is important information that needs to be included.

Conclusions support the results provided.

Reviewer 2 ·

Basic reporting

No comment

Experimental design

1. Isolation of EPCs. EPCs are known to be a heterogeneous population of cells. The authors use the uptake of acLDL and binding of UEA-1 as markers of EPCs. This should be further verified by expression of CD31 and VEGFR-2 (KDR).

2. Adenoviral mediated overexpression of Gab1. It is not clear why the authors need to overexpress Gab1 in the EPCs. Endogenous expression of Gab1 in EPCs is shown in Fig.1I and Fig.4A. Indeed, Fig.4A appears to show hardly any increase in Gab1 in the Ad-Gab1 cells compared with the untransfected cells. Fig.1G is confusing as it appears that a MOI of 20 is chosen based on cell proliferation rather than expression of Gab1. It seems strange that this effect on proliferation is not dose-dependent as a MOI of 24 appears to show no increase in proliferation?

The reliance on overexpression of Gab1 to generate results is a major flaw in the study. It would be better to study the effect of manipulating endogenous Gab1 in the EPCs, using siRNA mediated gene silencing, as an initial strategy before progressing to over expression of Gab1. Furthermore, overexpression of Gab1 appears to stimulate only a very small increase in HGF-stimulated proliferation (Fig.2A) and HGF-stimulated migration (Fig.3A) compared with HGF stimulation of control transfected cells? What about using more complex angiogenesis assays such as a tubular morphogenesis assay on a fibroblast monolayer (Richards M et al 2016 Methods Mol Biol 1430:159-166). This would allow the analysis of vascular structures with the EPCs.
It is hard to see what the clinical significance of this data is. Are the authors advocating overexpression of Gab1 in EPCs before infusion back into patients? What would be the advantage of this procedure considering the very small effects of Gab1 overexpression?

Validity of the findings

Overall, I think that the strategy employed to analyse Gab1 effects is flawed and not relevant to normal physiology. The authors should utilise a system where endogenous levels of the Gab1 protein can be utilised rather than an artificial system based on overexpression.

---

## Round 0.2 · accepted · Accept

Your article is now accepted, but please check grammar and writing style carefully while in production.

# ·

Basic reporting

The required changes have been made and are acceptable to improve the clarity of the paper.

Experimental design

Inappropriate methodological statements have been removed as requested and the descriptions are now clear enough to allow replication.

However, it is unclear how much effect Gab1 overexpression has on other signalling pathways. This should be investigated further to determine the specificity of the proposed method. Whilst the results obtained by the authors shows promise for increasing proliferation, the wider context must be considered.

Validity of the findings

The findings provide some additional, albeit incremental, advance in our current understanding and conclusions are reasonable.

Reviewer 2 ·

Basic reporting

No comment

Experimental design

No comment

Validity of the findings

Manuscript has been improved

Additional comments

I believe that the authors have made a number of revisions to the manuscript to improve the data presented and that this manuscript is now suitable for publication.